# Pop-out effects revisited: Within-array category pop-out and novel pop-out effects with picture stimuli

**John D. McCarthy, Phil Reed** ⓘ *

School of Psychology, Swansea University, Swansea, United Kingdom

* p.reed@swansea.ac.uk

## Abstract

Pop-out effects occur when a novel or different stimulus is presented in the context of an array of otherwise familiar or similar stimuli. The effect has been studied using words extensively, but little evidence exists for humans relating to nonverbal stimuli. Although the finding has implications for understanding features of stimuli that capture attention, contradictory findings exist, and previous paradigms have limited applicability to real world situations. Given this, an experiment employed a novel procedure to investigate whether category pop-out effects, where one item is drawn from a different category to the others, could be obtained with pictorial stimulus array. It also investigated whether pop-out effects could be generated with a single continuous pre-exposure procedure, as would be experienced in a naturalistic setting, or whether they were the results of biases introduced by the repetitive pre-exposure procedures, typically used in such studies. The latter finding would undermine the use of such findings to support ecologically-valid models of attention. Both of these questions were answered in the affirmative: category pop-out effects were obtained using picture stimuli; and such effects were obtained with a single continuous pre-exposure. Further development this novel procedure may allow exploration of evolutionary and neurological aspects of selective attention effects.

## Introduction

Learning which stimuli in the environment are important is a critical feature of adaptation, and such selective attention effects are noted across a wide range of procedures and species [1–3]. Selective attention refers to when one stimulus from a range of present cues controls behaviour. Understanding the features of stimuli that differentially capture attention and come to control behaviour have important implications for a variety of applied settings including air traffic control [4–6]. One set of phenomena that are important examples of such attentional capture are 'pop-out effects', which refer to situations in which unexpected or unfamiliar stimuli elicit strong behavioural control and attentional responses when presented in the context of expected or familiar stimuli [2, 7, 8]. Pop-out effects are important as they provide evidence relevant to suggestions that attentional responses are driven by certain aspects of the stimulus

**Data Availability Statement:** Anonymous data is stored securely, and is available on request form the corresponding author, or from the Head of the

University Psychology Ethics Committee Dr. Gabi
Jiga-Boy (g.jiga-boy@swansea.ac.uk).

**Funding:** The author(s) received no specific
funding for this work.

**Competing interests:** The authors have declared
that no competing interests exist.

array [8, 9]. In addition to many real-world applications, pop-out effects also have been
employed to explore differences in attentional processes across clinical populations [10, 11],
and have implications for understanding key learning phenomena [2, 3, 11].

Pop-out effects can be related to a range of stimulus attributes: relative novelty, where a
rarely-before-seen cue is presented among more familiar cues ('novel pop-out' [7]); or categor-
ical difference, where a lone item from one category is presented among several items from
another ('category pop-out' [12]). However, procedural differences between novel pop-put
and category pop-out studies typically exist and may underly some discordant findings [13,
14], making theoretical integration difficult. Firstly, the nature of the prompt specifying the
target cue to be searched for often differs between novel and category pop-out studies (post-
exposure or pre-exposure, respectively); secondly, the nature of the stimuli used can differ
(words versus pictures); and thirdly, the nature of the familiarisation procedure is often differ-
ent (intra or extra experimental).

The current study focuses on these three issues, and specifically examines the existence of
categorical pop-out effects with picture stimuli, using an intra-experimental familiarisation
procedure more typically employed for novel pop-out effects using word stimuli. As it is cur-
rently unclear whether similar factors are at play in novel and category pop-out effects [7, 13],
exploration of this procedure would help integrate theoretical explanations.

## Literature review

As noted above, a wide variety of experimental paradigms have been employed to study pop-
out effects, often confounding interpretations of the cross-experimental effects. For example,
pop-out can be manipulated by varying either novelty or categories of items; using either
words or symbols/pictures; using either pre- or post-exposure cueing to elicit identification of
the novel item; and altering familiarity by manipulations within the experiment versus relying
on previously established patterns of familiarity between stimuli. Moreover, many of these
procedures are conducted using techniques that reduce the level of ecological validity of the
task.

## Novel versus category pop-out

In the original novel pop-out studies [7], three types of word arrays were presented: all-familiar
arrays containing words that had been repeatedly presented together throughout the experi-
ment; all-novel arrays containing words never presented previously in the experiment; and
mixed-arrays composed of one novel word among familiar words. At test, participants were
asked to remember where a particular word stimulus had appeared in a spatial array. Mixed-
arrays that contained some novel and some familiar words produced a location-identification
advantage for novel over familiar items [8, 15–17]. Additionally, when a familiar word is pre-
sented against a background of other familiar words with which it has never been presented
previously, but which had always presented together previously, the unexpected familiar word
'pops out' [18]. However, some have suggested that such effects cannot be attributed to novelty
per se as all words had been pre-exposed, just not together, but may depend on unitisation
(association) of words in the array, and 'odd pop-out' has been suggested as more appropriate
[18]. Thus, if a target 'odd' word is presented in an array of unitised-familiar words, the former
will be sampled in preference to the unitised words.

These forms of pop-out ('odd' or 'novel') can be contrasted with 'category pop-out' [12, 19,
20]. For example, it is easier to find a letter amongst digits, or vice versa, than a letter amongst
letters, or digit among digits. Similar to the explanation suggested for odd pop-out [7], it has
been suggested [21] that category pop-out may be explained by the existence of associative

connections amongst members of the same category. In the example of letters and digits, above, as the categories of digits and letters are relatively small and constrained, the associative connections amongst their members would be strong. Given this, any stimulus presented as a target (e.g., a letter) is likely to prime many other members of that category (other letters), which will then attract attention, and make searching for the specific target letter more difficult when presented in the context of other letters. In contrast, if the target (e.g., a letter) is presented amongst items from a different category (e.g. numbers), then the letter will appear to 'pop out', and the search will be relatively rapid.

## Pictorial stimuli

The parameters of category pop-out studies can be altered to use pictures rather than words [13]. This can serve to increase the ecological validity of studies. However, in some studies that have used pictorial stimuli, target detection decreases rather than increases when the target is incongruent (novel) with the rest of the presented scene (e.g., a sofa presented in a street scene). There have been few, if any novel/odd pop-out effects reported using pictorial stimuli. However, interpretation of the effects of pictorial stimuli in categorical pop-out are not straightforward, as these studies often involve several procedural differences relative to pop-out studies using words. Firstly, a pre-exposure cueing technique, involving verbal instructions to search for a particular object, rather than a post-exposure search-cueing prompt, is often adopted. Secondly, extra-experimental rather than intra-experimental familiarity manipulations is employed. The role of many of these procedural changes in these discordant results has not been explored.

## Cueing effects

Priming effects [7, 21] may explain the role of verbal pre-cueing to search for an item (e.g., 'look for a sofa' [13]). Evidence from the semantic priming literature suggests that such verbal instructions may semantically prime subsequent picture identification [22, 23]. If the presentation of the word 'sofa' in the instructions semantically primes other members of the same class of objects (e.g., furniture), then subsequent presentation of a street scene would be novel in the context of the currently active (furniture) representations. Given this novelty, the street scene would receive processing at the expense of the target stimulus (sofa), and identification of the target would be impeded.

   In contrast, studies using picture stimuli that do not use verbal pre-cueing about the identity of the target report outcomes similar to novel/odd pop-out studies using words. When line drawings of stereotypical scenes (e.g., city, farm, kitchen) comprising mainly expected objects, but also containing some unexpected objects (e.g., an octopus in the farmyard), are presented, duration of first looks are longer for the unexpected objects when participants were not instructed to search for those objects [14]. During subsequent recognition tests, participants rarely noticed expected objects that were missing or changed from the original presentation, but almost always detected missing or changed unexpected objects [14]. These results concord with those reported for words when pre-cueing techniques are not used [7]. These results suggest the sampling decrement for unexpected objects [13] may be due to the use of a pre-cuing procedure, implying a role for priming through associations established between the stimuli presented.

## Extra-experimental and intra-experimental familiarity

These considerations suggest an empirical distinction between two types of associations that may drive attentional processes: associations assumed to exist pre-experimentally between

array items (extra-experimental); and associations formed between array items during the experiment (intra-experimental). The results from pictorial category pop-out experiments [13, 14] suggest attention is modulated by extra-experimental associations, as observers were not familiarised with scenes during a specified pre-exposure as in the studies of novel/odd pop-out [7, 18], but pictures were only familiar to the extent that they were stereotypical representations of naturalistic objects. It is not possible to explain word pop-out effects through extra-experimental associations, as array compositions are randomised, and novel pop-out [7], and odd pop-out [18], are explained through intra-experimental associations.

## Summary

Although pre-cueing and extra-experimental versus intra-experimental associations may explain some discrepant findings, and help integrate the explanation of various pop-out effects, it is still the case that category pop-out effects have not been established for pictures when familiarisation is done intra-experimentally. It remains possible they are the result of the familiarisation procedure, and it remains unclear whether category pop-out would be observed without prior target item specification using a pre cuing technique. This is the prime focus of the current study. Clarifying the nature of pop-out effects, and the role of various procedural manipulations, may help understand some of the theoretical and applied implications of pop-out more clearly.

## Current study aims

The current experiment compared localisation performance for picture stimuli, in a category pop-out study, on displays that included semantic conflict with that observed on displays with both semantic and episodic conflict. The basic structure of the study consisted of a pre-exposure phase, an orienting phase, and a probe phase. In the probe phase, participants could be tested on the location of an item in the preceding orienting array. This orienting display could contain pictures between which there was semantic conflict. Such displays consisted of three pictures drawn from one category, and one picture drawn from the other category. The right panel of Fig 1 gives an example of an orienting display in which there are three pictures of

Pre-exposure                                                                         Orienting

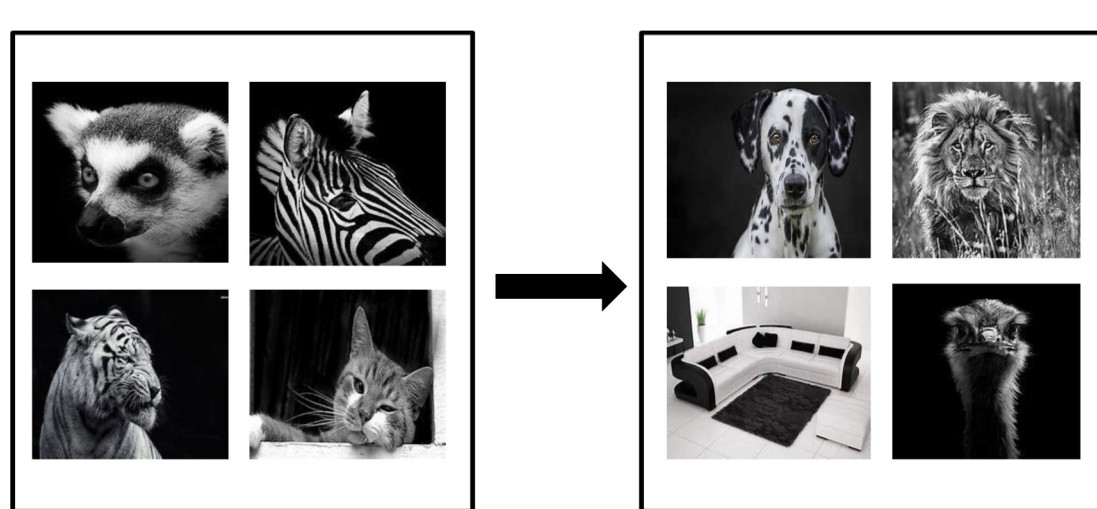

**Fig 1. Example of a pre-exposure array with four pictures from one category (animals), and an orienting display with three different pictures from that same category (animals), and one picture from a novel category (furniture).**

animals and one picture of a piece of furniture. It was predicted that priming effects would favour localisation of the lone category item (furniture) with just semantic conflict, despite the absence of prior instructions. Alternatively, the orienting display could contain both semantic and episodic conflict. The additional episodic conflict was introduced by prior exposure to the pictures drawn from the majority category item in the orienting display. For example, with simple semantic conflict, subjects might be tested on an orienting display containing three animal pictures and one furniture picture, as shown in the right panel of Fig 1. However, with both types of conflict, participants first would be pre-exposed to four pictures from one category (e.g., four pictures of animals, as shown in the left panel of Fig 1). They then would be tested with three different pictures from that category (animals) and one picture from a novel category (furniture), as shown in the right panel of Fig 1. Thus, in the orienting phase, three of the pictures would be from a familiar category, and one would be from a novel category. When the episodic effects were introduced, a novel pop-out effect was predicted that would further enhance performance on the one category item.

Out of a general concern for ecological validity, an additional difference between the present paradigm and previous studies, is the use of a single continuous pre-exposure, rather than several discrete pre-exposures to an array. The use of repetitive brief exposures is somewhat ecologically artificial, as in natural environments visual stimuli are usually continuously present, rather than repeatedly flashing in front of the subjects. The ecological importance of pop-out has been noted in the context of several studies using nonhumans [1, 2, 24]. Sampling items in an array under natural conditions may, therefore, follow a different course to that which occurs when repetitive stimulation is used. This presents the possibility that the pop-out effect is an artefact of imposing a sampling process, rather than a normal function of attention. Consequently, a single pre-exposure period was adopted in the present study as an additional test of the boundaries of pop-out phenomena.

## Method

### Ethical considerations

Ethical approval was given by the Ethics Committee of the University Psychology Department (Ref.: 1 2024 9545 8463). Data collection started on 13.4.24 and ended on 14.4.24. All participants gave their informed consent for the study, which was recorded in writing.

### Participants

20 undergraduate university students participated (14 female, 6 male), with a mean age of 20.8 years, participated for subject pool credit. G-Power calculations suggest that for 85% power, with a rejection criteria of $p < .05$, and a medium effect size ($f$ = .3), 19 participants would be needed for a one-way repeated-measures analysis of variance.

### Apparatus and stimuli

The experimental task was presented on a PC with a 40cm screen, and was written in Visual Basic 6.0. The stimuli consisted of 64 pictures drawn from a previously widely-used pool of picture stimuli [25]. These 64 pictures were drawn from eight different categories, with eight pictures in each category. The eight categories were divided into two sets: one set of four categories was used as 'pre-exposed' categories (kitchen utensils, clothing, tools, and musical instruments); and the other set of four categories was used as 'novel' (not pre-exposed) categories (transport, toys, furniture, and weapons). Examples of pictures drawn from the transport, furniture, and musical instruments categories can be seen in Fig 2.

**Fig 2. Examples of the pictures drawn from Snodgrass and Vanderwart (1980).**

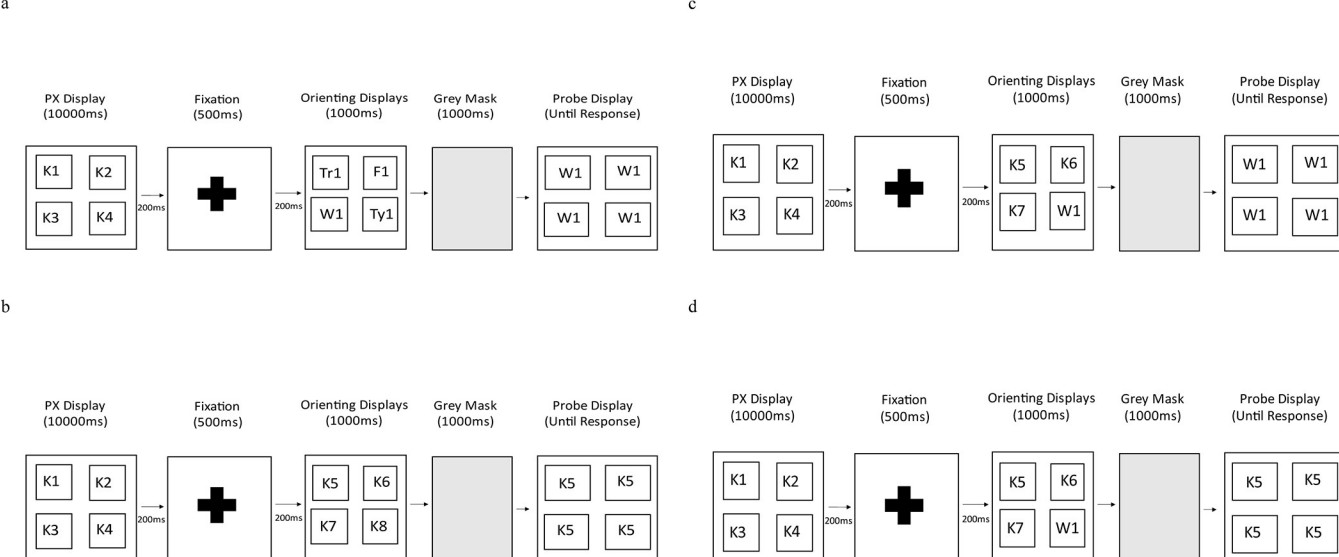

**Fig 3. Display sequences and timings for the conditions.** Fig 3A = all novel condition; Fig 3B = all-familiar condition; Fig 3C = one-novel condition; Fig 3D = three-familiar condition. K = kitchen utensil; Tr = transport; F = furniture; W = weapon; Ty = toy.

## Procedure

Participants were tested individually in quiet laboratory cubicles containing a desk, a chair, and a computer. Participants were seated about 50cm from the monitor, facing the monitor and with a computer keyboard in from of them.

The experiment adopted a within-subject design, with each participant being exposed to four conditions (all-novel, all-familiar; one-novel, or three-familiar). Each participant received 48 experimental trials, with trials from each condition presented in a random order. Each trial was split into three segments: pre-exposure, orienting, and probe test (see Fig 2). All subjects were informed of the display sequence prior to the experiment, and they were told that the initial pre-exposure pictures were irrelevant to the task.

The display sequence for these phases in each of the four conditions is represented in Fig 3. In each condition, arrays containing four stimuli were presented to the subjects. Each stimulus in an array subtended a visual angle of about 1.9˚ horizontally, and .64˚ vertically, from a viewing distance of 50cm. The entire array subtended an angle of about 5.1˚ horizontally, and 4.5˚ vertically.

Firstly, a 10,000ms pre-exposure segment was presented, in which participants were shown an array of four pictures all drawn from the same category as one another (e.g., kitchen utensils, clothing, tools, or musical instruments). This relatively long exposure was used to establish this category as familiar. This familiarisation array was followed on its' offset by a 500ms fixation point presented in the centre of the screen. Secondly, there was a 1000ms orienting segment comprising four stimuli. The four stimuli presented in this array were individually different from those presented in the preceding exposure-array. The four stimuli in the orienting display could be either: one from each of four categories assigned as novel (all novel); four new stimuli from the same category as presented during familiarisation (all familiar); or three new stimuli from the same category as presented in the preceding exposure-array, and one stimulus drawn from one of the novel categories. Whether this latter was in the one-novel or three-familiar condition depended on which stimulus was tested in the subsequent probe trial. This orienting-array was followed on its offset by a 1000ms grey mask filing the screen [7]. The

third segment of a trial was the probe test. In this, the same stimulus was presented in each of the four locations on the screen. The stimulus as the same as one of the stimuli presented in the preceding orienting-array. This display was presented until the participant made a response to one of four keys on the keyboard. These keys were marked so that each one corresponded to one of the four locations on the screen. The response was made to indicate the remembered location of the probed picture on the screen in the preceding orienting-display. This response produced an intertrial interval (ITI) of 2000ms. After this ITI, the next trial sequence began.

## All novel condition

There were 12 all-novel trials in which each subject received pre-exposure to four stimuli from one category of pictures from one of the pre-exposed sets; for example, they were exposed to four pictures from the kitchen utensils, or clothing, or tools, or musical instruments categories. Fig 3A shows an array where four stimuli from the kitchen utensil set were exposed. Across the 12 trials in this condition, there were three in which each of the four pre-exposed categories were used. The actual pictures used on each trial from that category were randomly selected from the pool available. This was followed by an orienting array comprising four pictures, one picture randomly selected from each of the novel sets (transport, toys, furniture, and weapons). Participants were subsequently probed for their location memory of a picture in the orienting array. A picture from each of the four novel categories was chosen as the probe on three of the 12 trials. In the probe trial, four images of the same stimulus were displayed on the screen (e.g., the picture from the weapon set that had been presented in the orienting display); one picture in each of the four locations. The participant had to indicate where the probe picture had appeared in the orienting display. Subjects indicated their response by pressing a key on the keyboard marked to correspond to one of the locations. This response was recorded and started the ITI prior to the next trial.

## All familiar condition

There were 12 all-familiar trials, during which each subject received pre-exposure to four randomly selected pictures from one the pre-exposed categories (kitchen utensils, or clothing, or tools, or musical instruments). Fig 3B shows four images from the kitchen utensil array. This pre-exposure array was followed by the orienting array that comprised another different four pictures drawn from that same set (shown in Fig 3B as four further pictures from the kitchen utensil set). The subsequent probe trial presented one of the pictures from the orienting display in all the four locations. Each category was chosen for test three times.

## One novel condition

For these 6 trials, the pre-exposure display comprised four randomly selected pictures from one of the categories used for pre-exposed stimuli (i.e. kitchen utensils, or clothing, or tools, or musical instruments). The orienting array was composed of three different pictures drawn from the same category as used in the preceding pre-exposure display (i.e. if kitchen utensils had been used in the pre-exposure, then three different kitchen utensils were shown in the orienting phase), and one randomly selected picture from one of the novel categories (transport, toys, furniture, and weapons). Fig 3C shows one picture from the weapon array being used. Each pre-exposed category was used three times, and each novel category was used three times. Participants were probed for the location of the one novel item from the orienting display (in Fig 3C they are tested on the weapon image).

### Three familiar condition

On the remaining 18 trials, participants received the same types of pre-exposure and orienting arrays as described for the one novel condition. However, they were probed for the location of one of the three familiar items from the orienting display (in Fig 3D, they are tested on one of the kitchen utensils).

### Results

The reports of correctly localised items in each condition were calculated, see Fig 4, and analysed by repeated-measures analysis of variance (ANOVA). Examination of Fig 4 suggests clear differences between localisation performance of the four array conditions, $F(3,57) = 10.35$, $p < .01$, $\eta^2_p = .352[.134:.487]$. A breakdown of this effect suggested that localisation of a novel picture is enhanced when it is presented in the context of a familiarised set of pictures. In other words, a lone novel picture amongst three familiar pictures is localised better than a novel picture among three novel pictures. This description was confirmed by a post hoc test, $F(1,19) = 9.55$, $p < .01$, $\eta^2_p = .335 [.119:.471]$. The data also show the familiar pictures were localised more accurately in an all-familiar array than in mixed-arrays, and a post hoc test revealed a significant difference between these performances, $F(1,19) = 16.40$, $p < .01$, $\eta^2_p = .463[.246:.582]$. There was also a localisation advantage for novel pictures in a mixed-array relative to the localisation familiar pictures in mixed-array. This result was confirmed with a post hoc test, $F(1,19) = 5.91$, $p < .05$, $\eta^2_p = .237[.045:.380]$.

### Discussion

The present experiment investigated whether category pop-out effects could be obtained with stimulus arrays composed of pictures, as there appeared to be some contradictory findings in

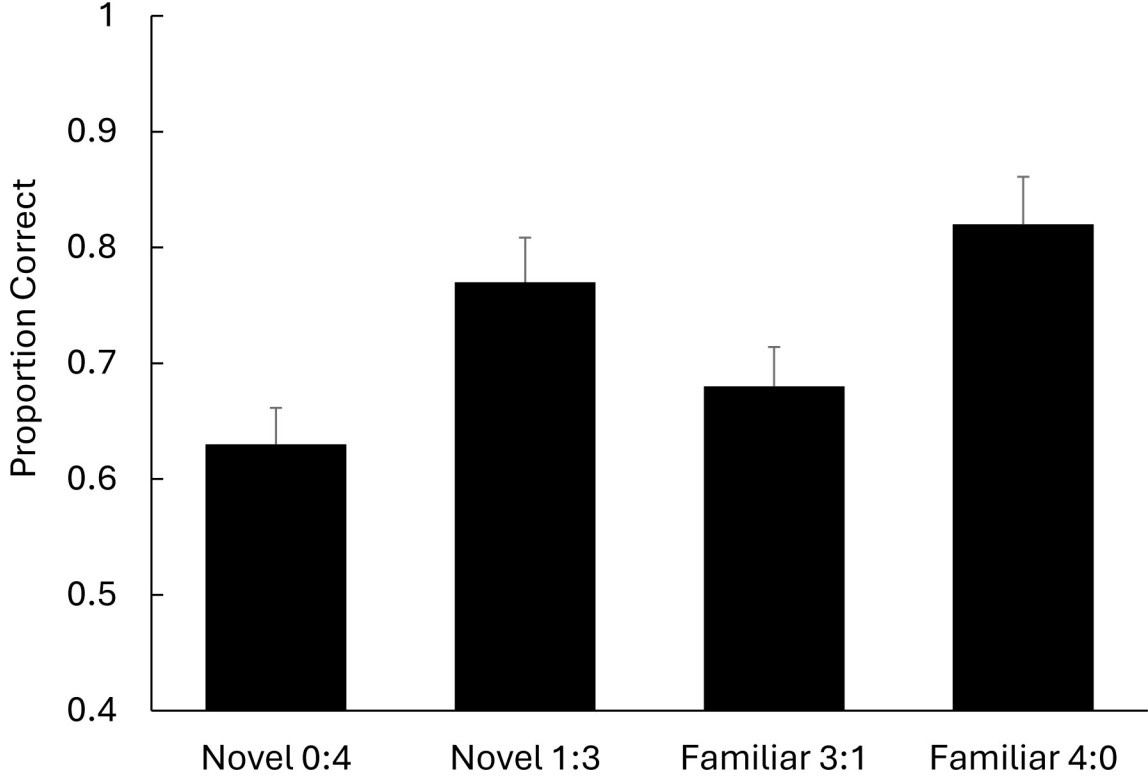

**Fig 4. Proportion of correct word localisations as a function of array type.** Error bars = standard errors.

the literature [13, 14]. Additionally, it investigated whether pop-out effects could be generated with a single continuous pre-exposure procedure, as would be experienced in a naturalistic setting, or whether they were the results of biases introduced by the repetitive preexposure procedures, typically used in such studies. These questions were answered in the affirmative; both category and novel pop-out effects were obtained using picture stimuli, and such effects were obtained with a single continuous pre-exposure.

The presently observed all-familiar baseline advantage has also been noted for word arrays [7]. The advantage noted to novel words in mixed-arrays, over those in all-novel arrays, is usually attributed to the redistribution of sampling towards the novel picture. Again, this difference in localisation performance can be attributed to a redistribution of sampling towards the novel picture, that produces a cost for familiar pictures, in the mixed-arrays. By contrast, when all pictures in the array are familiar, there is no attentional bias away from any of the array pictures, resulting in superior localisation performance for these familiar pictures relative to those familiar pictures in the mixed-arrays. Therefore, on the grounds of parsimony, both sets of effects can be attributed to a sampling redistribution towards the novel picture in mixed-arrays.

The results illustrated sampling can be biased by intra-experimental associations. When subjects were briefly presented with three pictures that had been pre-exposed together, and a lone novel picture, localisation performance showed a sampling bias towards the lone novel item. The finding of a pop-out effect for display durations of only 50ms replicates, with pictures, the pop-out obtained with brief word displays [7]. In a previous exploration of novel pop-out [18], novel pop-out was observed with displays ranging from 33ms to 200ms. The differentiation of stimuli with such brief exposure suggests that a novel pop-out effect is a reflection of a bias in initial sampling behaviour (i.e. attention capture by novel items), rather than effects that occur after an object is sampled (e.g., novel-lingering). According to estimates of attentional switching time, display durations of only 50ms should only allow one object to be attended in each orienting array [26], meaning that any localisation differences are likely to reflect differences in how easily the stimuli initially capture attention on their presentation.

The current experiments supported the previous findings concerning pop-out effects with pictures [14], albeit using a radically different paradigm. In doing so, the present studies did not corroborate the conclusions drawn on the basis of failures to show categorical pop-out with pictures [13]. Of course, there are procedural differences between the current study and the latter report. Two differences have been drawn out in the General Introduction to this report: the lack of an explicit familiarisation procedure; and the use of a directed attentional procedure [13]. Although not explicitly tested, it seems more likely that it is the latter of these differences, rather than the lack of pre-exposure, that is responsible for the differences obtained in the results in the present experiment and those reported previously [13]. This suggests that the use of a directed attention task [13] may be responsible for the differing results obtained.

The use of single pre-exposure period in the present study may allow closure integration of such human popout effects with those noted for nonhumans [1, 2, 24]. Selective attentional mechanisms like novel popout may be an evolutionary adaption for managing attentional resources, and developing a procedure for humans that is similar to those used for nonhumans specially in ecologically valid settings, may allow easier comparisons between the results from different species. In any case, that similar results were noted with a continuous pre-exposure to those previously found with repeated exposures suggests that this is not factor in modifying the effects.

## Limitations and future studies

In the current study, it should be acknowledged that the same assignment of picture categories to pre-exposed and novel conditions were used for all subjects. It is possible that some categories could be intrinsically more likely to capture attention than others. Future studies could consider counterbalancing the assignment of picture categories to these conditions across subjects. Additionally, further experiments that vary the parameters of the stimulus (e.g., number of targets and distractors, arrangement of the stimuli in grids or randomly, different colours) may serve to extend the generality of the findings, and make even closer contact with more ecologically valid experimental settings.

It might be considered that category pop-out and novel pop-out are, on some levels, necessarily confounded in this design. In the one-novel condition, the single novel item was also the only item from one category among items from a different category (i.e. a category singleton). One suggestion is that higher accuracy for this condition compared to the three-familiar and all-novel conditions might be explained by either category pop-out or novel pop-out (or a combination of both). However, in terms of the orienting display, all the items are novel in the sense that none of them appeared in the preceding pre-exposure display; or, alternatively, all items may be regarded as familiar, as they were all exposed previously in other displays in this study. They are novel in the sense that they represent a categorical change, suggesting categorical pop-out drives the effect. Nevertheless, future studies could explore pure category pop-out by comparing two conditions which are the same as the one-novel and all-familiar conditions, but either without pre-exposure, or with pre-exposure to a category which does not appear in the orienting display (as in the all-novel condition).

Gaining a greater understanding of the natures of stimuli that capture attention, and pop-out from a background, has implications for a range of applied areas including design composition, marketing, advertising, gaming design [4–6]. The findings emerging from the current study suggest parallels with notions of the Visual Hierarchy from Gestalt theory and established perceptual organisation principles. The key concept being that 'isolated objects predictably stand-out, colour and shape being standard examples, in more abstract compositions 'continuation' and 'closure' contribute to usability and decision making [27]. The use of the more ecologically-valid procedure outlined here may serve to forward development of these applications and links to theories from different backgrounds.

## Conclusion

The present experiment successfully demonstrated that novel pop-out and category pop-out effects can be obtained with stimulus arrays composed of pictures. Moreover, the present series demonstrated that such pop-out effects could be generated with a single continuous pre-exposure procedure, as would be experienced in a naturalistic setting, rather than being the result of biases introduced by the repetitive preexposure procedures typically used in such studies. The ecological importance of popout has been noted in the context of several studies using nonhumans. The development of such a procedure may allow closer contact with studies from nonhumans that have also explored the evolutionary and neurological aspects of selective attention effects.

## Author Contributions

**Conceptualization:** John D. McCarthy, Phil Reed.

**Data curation:** John D. McCarthy.

**Formal analysis:** John D. McCarthy.

**Methodology:** John D. McCarthy, Phil Reed.

**Resources:** Phil Reed.

**Software:** John D. McCarthy.

**Visualization:** Phil Reed.

**Writing – original draft:** John D. McCarthy, Phil Reed.

**Writing – review & editing:** John D. McCarthy, Phil Reed.

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
