## [Decision Letter · Decision Letter 0]

2 Jul 2024

PONE-D-24-15452Pop-out effects revisited: Within-array category pop-out and novel pop-out effects with picture stimuliPLOS ONE

Dear Dr. Reed,

Thank you for submitting your manuscript to PLOS ONE. After careful consideration, we feel that it has merit but does not fully meet PLOS ONE’s publication criteria as it currently stands. Therefore, we invite you to submit a revised version of the manuscript that addresses the points raised during the review process.

We look forward to receiving your revised manuscript.

Kind regards,

Hosam Al-Samarraie

Academic Editor

PLOS ONE

2. In this instance it seems there may be acceptable restrictions in place that prevent the public sharing of your minimal data. However, in line with our goal of ensuring long-term data availability to all interested researchers, PLOS’ Data Policy states that authors cannot be the sole named individuals responsible for ensuring data access (http://journals.plos.org/plosone/s/data-availability#loc-acceptable-data-sharing-methods).

Additional Editor Comments:

The study of pop-out effects offers some interesting findings. However, I still believe there is room for improvement. The reviewer outlines some relevant points that I hope you and your co-authors can address. I also have some concerns that you can address in a revised version. For example:

1. Your manuscript can benefit from using a clear structure for the introduction and method. As it stands, there is a mix between previous studies and the gap that should be, to some extent, covered in the literature review section. Therefore, I would like you to consider separating and expanding on previous studies in the introduction and putting them into one separate section.

2. It would be good to discuss attentional bias in relation to pop-out effects in the literature review section. This is mainly due to the nature of your experiment and results. It is unclear what measure was used in this study to answer the research questions.

3. The apparatus section should provide more details about the technology used, conditions, and settings.

4. The discussion section is well written but lacks structure. To improve its readability, I would suggest that you take out and expand on the limitations and implications into a separate section.

5. I found the following statement a bit confusing: “Therefore, the biases observed in this study most probably reflect differences in how this initial selective process occurs.” Can you please elaborate and cite relevant sources here?

Reviewers' comments:

Reviewer's Responses to Questions

**Comments to the Author**

1. Is the manuscript technically sound, and do the data support the conclusions?

Reviewer #1: Partly

2. Has the statistical analysis been performed appropriately and rigorously? 

Reviewer #1: I Don't Know

3. Have the authors made all data underlying the findings in their manuscript fully available?

Reviewer #1: Yes

4. Is the manuscript presented in an intelligible fashion and written in standard English?

Reviewer #1: Yes

5. Review Comments to the Author

Reviewer #1: Pop-out effects revisited: Within-array category pop-out and novel pop-out effects with picture stimuli

The Abstracts indicate the work is novel in that it investigates ‘uncertainties’ in the current experimental work; affirmative conclusions are suggested regarding the effect of pre-exposure however this not easy to follow in terms of picture stimuli and a clear problem context.

The introduction offers broader contextual value of the work but greater insight into the implications of the research question could help more clearly position the potential of the work in the mind of a reader new to the field. Other studies cite design examples such as air traffic control displays and interface design, a context to help identify impact.

The literature review is sound, well written and concise but to support clarity for a wider audience the numerous experimental paradigms and stimuli arrays could be perhaps summarised for the benefit of those not fully conversant with the core literature.

The top-level explanations and examples are readily accessible ‘a letter amongst digits’ or the ‘sofa in a street scene etc., the addition of visual information examples could resolve some issues relating to contextual understanding. A more critical positioning of the key theories and their application again could also support added value for the reader, in terms of problem understanding.

The visual stimuli, line-drawings of familiar scenes inserted with unfamiliar/novel objects, likely to reality ‘Pop Out’ seems intuitive. The study framework described for ‘localized performance’ with semantic and episodic conflict in the three experimental phases would be enhanced with supporting visual information more clarity embedded in the text.

The use of standard illustrating is described in the methodology signposted, for figure-1 but as the description continues the detail of the methodology are not revealed for the none initiated e.g., the difference between a measure of and implication for 10,000ms, 500ms and 100ms, grey-mask filing, probe test and probe picture. Inference is evidenced but not explicit, problematic for repeatability but detrimental to following the later discussion.

The work is intriguing and could add value for those who investigate visual response for other disciplines, design composition, marketing, advertising, gaming design, eye-tracking based research and perhaps Virtual Reality.

The work seems parallel to understanding of Visual Hierarchy founded in Gestalt theory and established perceptual organisation principles. The key concept being that ‘isolated objects predictably stand-out, colour and shape being standard examples, in more abstract compositions ‘continuation’ and ‘closure’ contribute to usability and decision making.

Cleary the authors expertise is not in question, their underpinning understanding of the field of research, however without prior knowledge of core literature the work is simply hard to follow. For the work to be published the authors need to consider clarifying the research problem, positioning the value of the research and it potential value. Assumption regarding what might seem obvious or basic to those with insider knowledge needs to be decoded, made accessible for a wider readership.

6. PLOS authors have the option to publish the peer review history of their article (what does this mean?). If published, this will include your full peer review and any attached files.

Reviewer #1: No

---

## [Author Response · Author response to Decision Letter 0]

20 Jul 2024

Reviewer’s comments:

The Abstracts indicate the work is novel in that it investigates ‘uncertainties’ in the current experimental work; affirmative conclusions are suggested regarding the effect of pre-exposure however this not easy to follow in terms of picture stimuli and a clear problem context.

We have expanded the Abstract to give a clearer problem statement, and to better contextualise the findings. 

The introduction offers broader contextual value of the work but greater insight into the implications of the research question could help more clearly position the potential of the work in the mind of a reader new to the field. 

We have mentioned the broader implications of pop-out effects for a range of fields, such as air traffic control designs and marketing in the Introduction to set the work in a broader context for readers.

The literature review is sound, well written and concise but to support clarity for a wider audience the numerous experimental paradigms and stimuli arrays could be perhaps summarised for the benefit of those not fully conversant with the core literature.

We have summarised the range of key procedural differences across studies to be addressed at the start of the new Literature Review section, and have sought to structure this new section into sub-sections discussing the various aspects of the literature that are important. 

The top-level explanations and examples are readily accessible ‘a letter amongst digits’ or the ‘sofa in a street scene etc., the addition of visual information examples could resolve some issues relating to contextual understanding.

We have added a new figure which may help with the visualisation of the procedure. 

A more critical positioning of the key theories and their application again could also support added value for the reader, in terms of problem understanding.

We have noted some of the key theories in the initial introduction to better contextualise the current findings, and we have referred to these throughout. However, we are mindful of the need not to over-interpret these data.

The study framework described for ‘localized performance’ with semantic and episodic conflict in the three experimental phases would be enhanced with supporting visual information more clarity embedded in the text.

We have included a figure in the Introduction with an example of the types of displays that are discussed in the text. 

The detail of the methodology are not revealed for the none initiated e.g., the difference between a measure of and implication for 10,000ms, 500ms and 100ms, grey-mask filing, probe test and probe picture. Inference is evidenced but not explicit, problematic for repeatability but detrimental to following the later discussion.

We had added more detail about the procedure in the Method section, and we hope that this clarifies the sequence of events.

The work is intriguing and could add value for those who investigate visual response for other disciplines, design composition, marketing, advertising, gaming design, eye-tracking based research and perhaps Virtual Reality.

We have added mention of the fields that these effects may be relevant for in the Discussion section.

The work seems parallel to understanding of Visual Hierarchy founded in Gestalt theory and established perceptual organisation principles. 

We have mentioned the parallels with these views in the Discussion, and noted that the current ecologically-valid design may help to further links to this literature and applied fields such as usability and decision making.

---

## [Decision Letter · Decision Letter 1]

28 Aug 2024

Pop-out effects revisited: Within-array category pop-out and novel pop-out effects with picture stimuli

PONE-D-24-15452R1

Dear Dr. Reed,

We’re pleased to inform you that your manuscript has been judged scientifically suitable for publication and will be formally accepted for publication once it meets all outstanding technical requirements.

Kind regards,

Hosam Al-Samarraie

Academic Editor

PLOS ONE

Additional Editor Comments (optional):

Thank you for addressing all the comments and suggestions.

Reviewers' comments:

Reviewer's Responses to Questions

**Comments to the Author**

1. If the authors have adequately addressed your comments raised in a previous round of review and you feel that this manuscript is now acceptable for publication, you may indicate that here to bypass the “Comments to the Author” section, enter your conflict of interest statement in the “Confidential to Editor” section, and submit your "Accept" recommendation.

Reviewer #1: All comments have been addressed

2. Is the manuscript technically sound, and do the data support the conclusions?

Reviewer #1: Yes

3. Has the statistical analysis been performed appropriately and rigorously? 

Reviewer #1: I Don't Know

4. Have the authors made all data underlying the findings in their manuscript fully available?

Reviewer #1: Yes

5. Is the manuscript presented in an intelligible fashion and written in standard English?

Reviewer #1: Yes

6. Review Comments to the Author

Reviewer #1: It is pleasing to see that with additional contextualisation and explanation for the none experts the paper has greater opportunity for impact amongst a wider audience particularly those with similar research addenda’s in different discipline areas.

7. PLOS authors have the option to publish the peer review history of their article (what does this mean?). If published, this will include your full peer review and any attached files.

Reviewer #1: No

---

## [Editor Report · Acceptance letter]

10 Oct 2024

PONE-D-24-15452R1 

PLOS ONE

Dear Dr. Reed, 

I'm pleased to inform you that your manuscript has been deemed suitable for publication in PLOS ONE. Congratulations! Your manuscript is now being handed over to our production team.

Kind regards, 

on behalf of

Dr Hosam Al-Samarraie 

Academic Editor

PLOS ONE